# Measuring health financing vulnerability due to reductions in official development assistance: A conceptual framework with empirical application across 47 African countries

**James Avoka Asamani**[1,2]*, **Sophie Faye**[1], **Kouadjo San Boris Bediakon**[1], **Hilary Kipruto**[1], **Sunny C. Okoroafor**[1], **Janet Kayita**[1], **Ame Dioka**[1], **Azzam Ali**[1], **Benson Droti**[1], **Ogochukwu Chukwujekwu**[1]

1 Health Systems and Services Cluster, WHO Regional Office for Africa, Brazzaville, Congo, 2 Centre for Health Professions Education, Faculty of Health Science, North-West University, Potchefstroom, South Africa

* Asamanij@who.int

## Abstract

Despite a 33% increase in Official Development Assistance (ODA) from 2019 to 2023, aid declined by 7% in 2024 and projected to further decline in 2025–2027. Health systems in Africa are being impacted by the sudden aid freezes and cuts. There are no multicriteria metrics to estimate the vulnerability of countries to facilitate dialogue and decisions. This paper proposes an approach to assess country's vulnerability to external aid cuts, considering dynamic health financing and macro-fiscal risk factors. We proposed a framework using four proxy health financing and macro-fiscal factors. Using a polychoric principal component analysis to derive the weight of contribution of each variable, we parameterized the framework with data from existing databases. We applied the framework to assess the health financing vulnerability to external aid cuts/freezes across different countries and explored its validity. Health expenditure per capita in the WHO African Region averaged US$125.2 of which almost 24% depended on external funding. Almost 70% of countries face potential stagnation or contraction in public spending, while 49% of the countries are in debt distress or at high risk of debt distress. The interplay among these factors resulted in the vulnerability of countries averaging almost 60% (range: 23 – 92%), with 27 countries facing "very highly" or "highly" vulnerable health financing situations. Health financing vulnerability explained 31.4% of the variation in UHC attainments and 32.8% of the variation in the number of people pushed further into poverty. We conclude that low spending on health, high donor dependence and unfavourable macro-fiscal conditions, including debt crisis is driving heightened vulnerability across Africa. Countries need to protect essential health services by prioritizing the most vulnerable populations, protect their health budgets through increasing efficiency, generating new revenue through taxes, aligning external aid with national plans and priorities, and better pooling health spending.

**Data availability statement:** Yes - all data are fully available without restriction; All relevant data are within the paper and its Supporting information files.

**Funding:** The authors received no specific funding for this work.

**Competing interests:** The authors have declared that no competing interests exist.

## 1. Introduction

Health financing is the foundation for ensuring adequate and sustained investments in various aspects of health systems, translating inputs into health service access and, ultimately, health outcomes [1,2]. Previous analysis in the African Region established the linkage and elasticity between health expenditure per capita and critical inputs for health service delivery/coverage such as health workforce, medicines availability, infrastructure, and diagnostic readiness that all play a role in the level of universal health coverage (UHC) attainment of countries [3,4]. In 2019, it was estimated that investing in health will save $2.9 trillion by 2030 in terms of mitigating productivity losses annually due to premature mortality and preventable, early onset morbidity [5].

In 2001, Member States of the African Union adopted the Abuja Declaration, committing to invest at least 15% of their annual national budget in health. Despite this political commitment, only a few countries have met this target [6–8], and average health spending from national budgets has remained stagnant at about 7% over the last two decades, reflecting a lower than promised prioritization of health in country budgets [6], as shown in Fig 1. Only three countries have met the Abuja target. To put in context, globally, countries spent an average of 10.8% of general government expenditure on health, which only seven (7) countries in the African region have reached or surpassed this global average. Additionally, almost 60% of countries in the African region allocated less than 7% of their national government spending to health.

African countries have relied on external financial support from overseas governments and development agencies for an average of 24% of health spending, but in some countries, the external funding contributes 45% or more (up to 65%) to their health spending [6,9]. Following a 33% global increase in the volume of Official Development Assistance (ODA) from 2019 to 2023, it recorded a decline of 7% in 2024 [10], and since the beginning of 2025, there has been a sudden reduction of donor funds [10–12] due to global fiscal constraints, shifting donor priorities, changing geopolitical dynamics and competing emergencies [13,14]. In particular, the United States announced a cancelation of 83% of aid-related contracts in March 2025, which resulted in halting critical health programs across Africa and beyond [15]. A survey conducted by the OECD suggests that development aid could face further reductions of between 9% and 17% from 2024 level through to 2027. Other analyses suggest that aid levels in 2025 could decline by as much as 40% relative to the 2023 baseline [12]. According to a recent assessment by WHO across 108 countries, the adverse impact of these precipitated donor transitions seems to be dire – potentially at 75% of the level of disruption to health systems functioning seen during the height of the COVID-19 pandemic [16].

While several high-level analyses and anecdotal reports have highlighted the potential implications of aid freezes or cuts on African health systems, a comprehensive and systematic assessment remains absent. Specifically, there are no empirical assessments on the degree of vulnerability that different countries face in the context of such external financing shocks, and there are no known widely

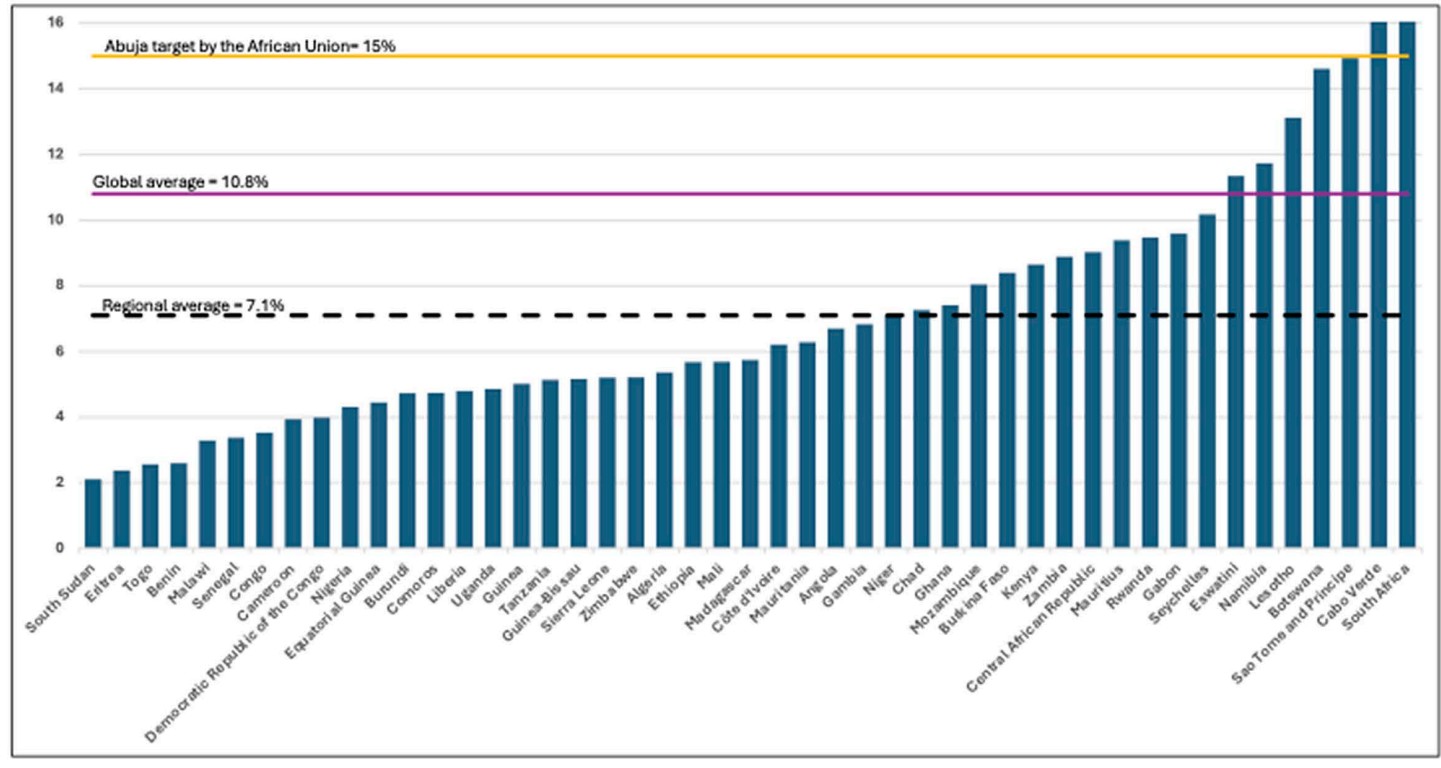

**Fig 1. Share of general government health expenditure as a percentage of total general government expenditure by country, 2022, WHO African Region. Data source: WHO's global health expenditure database.**

available methodological frameworks for such analyses. To address this methodological and evidence gap, this paper aims to put forward a proposed conceptual framework and an empirical application for assessing the extent of vulnerability of different countries' health financing systems to external aid cuts or freezes using four dynamic health financing and macro-fiscal risk factors acting in different directions. Based on the results, we have identified priority countries that should inform strategic responses and enhancing the resilience of health financing and overall health systems across the continent.

## 2. Methodology

### Conceptual framework and assumptions

We conducted a search for appropriate theoretical and/or analytical frameworks and found that there are a few frameworks and empirical studies that explored the level of vulnerability of households or countries to financial crises [17–19], and several others that demonstrates how health poverty alleviation projects could reduce catastrophic out-of-pocket spending and economic vulnerability of households [20–24]. However, we did no find known conceptual or empirical frameworks that examine countries' health financing vulnerability to sudden financial shocks (sudden freeze or withdrawal of funds). Nonetheless, there is a fairly theorized economic vulnerability index, defined as "… the risk of a (poor) country seeing its development hampered by natural and external shocks". In the computation of the economic vulnerability index, three key components are taken into account: (i) the size and frequency of the exogenous shocks, (ii) the duration of exposure to shocks and (iii) the capacity to respond to the shocks [25,26].

We utilised the concept of economic vulnerability described by Briguglio and Galea [25] and Guillaumont [26,27] to construct a health financing vulnerability score in which we considered the size of the shock to be analogous to degree of a foreign aid dependency by countries for their health spending. The duration of the shock was considered to be chronic since there are no expected timelines for restoration of the cut foreign aid. We considered the capacity to respond to the shock at three levels – (1) level of pre-existing health spending per capita, (2) ability to expand budget or budget space potential and, (3) capacity to borrow. We deemed each of these constructs as a factor of exposure to health financing vulnerability and assigned ordinally scaled values known as "levels of exposure" to each factor of exposure.

We posit that in response to reductions or cuts in external aid, countries would be compelled to make a range of policy and investment decisions. These may include reprogramming or reorganizing service delivery to operate within existing expenditure levels - assuming those levels are reasonably sufficient. Alternatively, governments may opt to increase domestic health spending to partially or fully compensate for the potential loss of donor support. The feasibility of this approach, however, depends on several factors: the available budget space, the extent of the gap to be filled due to external aid cut, and the government's capacity to borrow - either domestically or internationally - as a short- to medium-term mitigation strategy.

Drawing on the models proposed by Briguglio and Galea [25] and Guillaumont [26,27], we assumed that the health financing vulnerability of countries to the external financial shocks depends on four parsimonious factors: (1) current health spending per capita, (2) extent of dependency on foreign aid or external financing of health, (3) government's budget space potential, and (4) borrowing capacity. We deemed each of these constructs as a "*factor of exposure*" to health financing vulnerability and assigned ordinally scaled "***levels of exposure***" to each factor of exposure. Fig 2 illustrates the conceptual framework we developed based on these assumptions to assess the vulnerability of countries in the African Region to sudden or even progressive external funding freezes and cuts.

**Assigning levels of exposures to reflect countries' propensity to be vulnerable across the four factors**

**Current health expenditure per capita.** We assumed that countries with lower levels of current health expenditure per capita are expected to have an elevated risk of vulnerability if external aid is reduced or cut. In the African region,

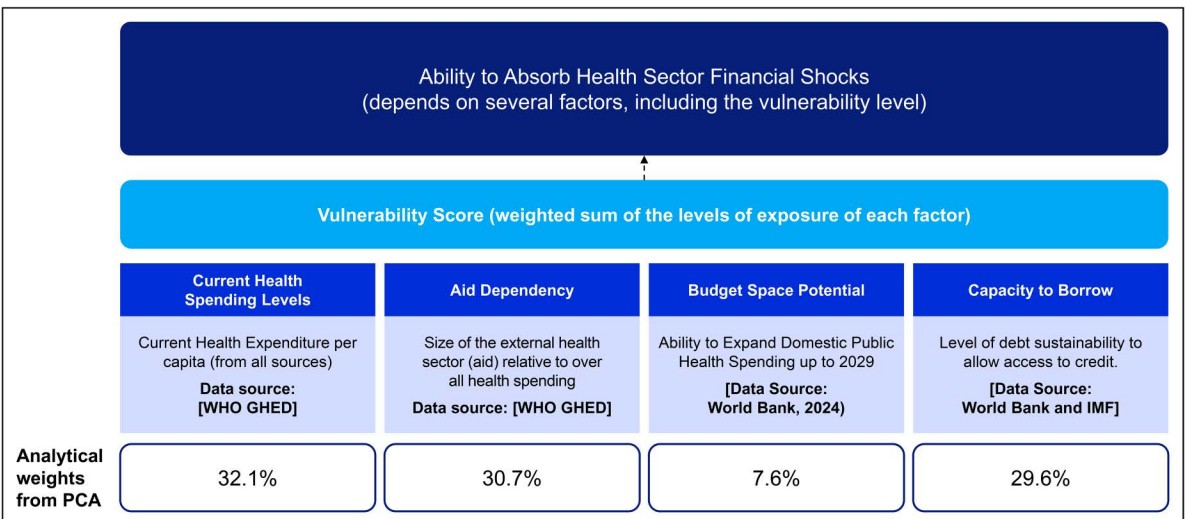

**Fig 2. Conceptual Framework for assessing the vulnerability of countries' Health Financing systems to sudden freezes or cuts in External Aid.**

the latest available data (as of 2022) indicates current health expenditure per capita ranges from less than US$30 to slightly over US$700 across 47 countries [6]. The countries were grouped into percentiles, with those in lower percentiles assigned higher levels of exposure to reflect relatively higher tendency of vulnerability, and those in higher percentiles assigned lower levels of exposure to reflect relatively lower tendency of vulnerability. The use of percentile cutoffs allowed for a standardized, data-driven approach to categorize countries relative to one another based on their distribution, enabling comparative vulnerability assessment while minimizing the influence of extreme outliers. Countries in the 20th percentile (less than US$55 per capita) were assigned a level of exposure of four; those in the 40th percentile (above US$55 but less than US$85 per capita) a level of exposure of three; those in the 60th percentile (above US$85 but less than US$180 per capita) a level of exposure of two; and those in the 80th percentile (more than 180 per capita) a level one exposure. Notably, countries in the 60th percentile had a per capita health spending of approximately $85, which almost coincides with the US$86 per capita spending considered as the minimum to make progress in delivering basic essential health packages [28]. The 20th, 40th, 60th, and 80th percentile cut points for the groupings were selected to allow a more granular differentiation of countries' per capita health expenditure levels than standard quartiles would permit. Using quartiles would have grouped together several countries with substantively different vulnerability profiles, thereby masking important heterogeneity. The chosen percentile thresholds therefore provide a more sensitive classification for analysing relative exposure.

**The extent of foreign aid dependence.** Linked to overall spending per capita is the proportion of health spending that comes from external sources or dependence on foreign financing. We assumed that the higher the proportion of a country's health spending funded by external sources (foreign aid), the more it is likely to suffer from financial shocks resulting from sudden or gradual reductions or cuts in foreign aid. Countries in the African region exhibit varying degrees of foreign aid dependency, ranging from nearly 0% in countries such as Seychelles, Mauritius, and Algeria to as high as 65% in other countries such as Malawi. To reflect this diversity, a five- level scale of exposure was used: countries with foreign aid dependency from 0-9% were assigned the lowest level of exposure of one; 10–14% a level of exposure of two; 15–25% a level three exposure; 25–44% a level four exposure; and those above 45% a level five exposure, indicating higher tendency of vulnerability.

**Budget space potential.** The ability of a country to fully or partially absorb financial shocks to the health sector caused by sudden or progressive foreign aid freezes, or cuts also depends on the extent of budget space potential. We defined budget space potential as the ability of the government to increase health spending in the short to medium term without compromising other key priorities [29]. According to its projections through 2029, the World Bank classifies countries into three categories: those with budget space expected to expand, remain stagnant, or contract [30]. In this analysis, countries whose budget space potential is expected to contract were assigned a higher level of exposure of three, those expected to stagnate a level two exposure, and those expected to expand a level one exposure.

**Capacity to borrow.** One of the short-term mitigation measures a country can explore when faced with sudden financial shocks to the health sector is borrowing from domestic and/or international sources. However, the capacity to borrow depends on the debt sustainability levels of the country – defined as "... the government [being] able to meet all its current and future payment obligations without exceptional financial assistance or going into default" [31]. Thus, a country with a sustainable debt can borrow additional funds to temporarily offset the loss of donor support. In contrast, for a country already experiencing or approaching debt distress (low debt sustainability), further borrowing is considered risky or potentially unviable. Regularly, the IMF and World Bank undertakes debt sustainability analysis (DSA) to guide borrowing and lending decisions [32]. In this analysis, therefore, we apply the IMF and World Bank's criteria for debt sustainability. Countries already in debt distress or at very high risk are considered more vulnerable and are assigned an exposure level four. Countries at high risk of debt distress receive an exposure level three, those at moderate risk are scored exposure level two, and countries with low or no risk – or those not classified – are assigned an exposure level one. The levels of exposure assigned to each country across the four variables are detailed in S1 File.

## Data sources

Data for this paper was systematically triangulated from various publicly available databases. Health expenditures and foreign aid dependency data were taken from the World Health Organization (WHO) Global Health Expenditure Database (GHED) [33]. GHED provides internationally comparable data on health expenditures from all sources of financing - governments, donors, insurance companies, households, etc. - for nearly all WHO Member States.

The budget space potential – a projection of government health spending was sourced from a World Bank publication on government health spending outlook [30], and the risk of debt distress (used in this paper as capacity to borrow) was taken from the World Bank and IMF debt sustainability analysis summary database [32,34].

## Deriving weights of contribution for each factor of exposure

To reflect real-world contributions, we assumed that each factor of exposure would affect health system vulnerability differently. Hence, equal weights without empirical or conceptual justification could misrepresent the nuanced roles of the factors and introduce bias and may overstate the role of less influential factors. We applied Polychoric Principal Component Analysis (PCA) to derive weights for the contributing factors of exposure for the vulnerability score. Traditionally, PCA is commonly used for reducing dimensionality and summarizing information from multiple variables, but it assumes that input data are continuous, have a linear relationship, and are measured on at least an interval scale [35]. These assumptions do not hold for the indicators used in this study, which are ordinal categorical variables [36]. Polychoric PCA was selected because it preserves interpretability in policy contexts where ordinal ratings - such as risk categories or financing levels - are commonly used, ensuring the resulting vulnerability scores remain relevant for decision-making.

The four indicators - Per Capita Current Health Expenditure (CHE), Foreign Aid Dependency, Budget Space Trajectory, and Risk of Overall Debt Distress - were scored on discrete ordinal scales. These scores reflect underlying latent traits (e.g., fiscal capacity, donor dependency) rather than direct numerical quantities. Using conventional Pearson-based PCA on such data can lead to misleading results due to violations of scale assumptions. Polychoric PCA, which replaces the correlation matrix with a polychoric correlation matrix, was considered more appropriate because it estimates the correlation between latent continuous variables that underlie observed ordinal indicators [37,38].

In terms of process, we began by computing the polychoric correlation matrix using the psych package in R. This matrix models each pair of the ordinal variables as if they are thresholds of underlying continuous distributions. PCA was then applied to the polychoric matrix to produce principal components that summarize the shared variance among the indicators. The derived weights are summarised in Table 1.

## Computing the vulnerability score from the factors of exposures

For each variable, the assigned level of exposure (ordinal score) was divided by the maximum attainable level of exposure and then multiplied by the empirical weight derived from the polychoric principal component analysis to derive the score for that component. The scores of the four components were then summed to yield the vulnerability score for each country.

**Table 1. Weights of the contribution of the factors of exposure to vulnerability.**

| Category | Weight (%) |
|---|---|
| CHE per capita Weight | 32.10 |
| Foreign Aid Dependency (EXT % of CHE) Weight | 30.70 |
| Budget Space Potential Weight | 7.60 |
| Capacity to Borrow | 29.60 |

The **vulnerability score** (**V**$_{Score}$) for each country was calculated as follows:

$$\text{VScore} = \left( \frac{\text{CHE}_{p.c}}{4} \times W_{che} \right) + \left( \frac{\text{FAD}}{5} \times W_{fad} \right) + \left( \frac{\text{BSP}}{3} \times W_{bsp} \right) + \left( \frac{\text{CTB}}{4} \times W_{ctb} \right)$$

(1)

***Where:***

- CHE$_{pc}$: Level of exposure from a country's Current Health Expenditure per capita.

- W$_{che}$: Weight assigned to the contribution of Current Health Expenditure per capita to the country's overall vulnerability score.

- FAD: Level of exposure from a country's Foreign Aid Dependency

- W$_{fad}$: Weight assigned to the contribution of Foreign Aid Dependency to the country's overall vulnerability score.

- BSP: Level of exposure from a country's Budget Space Potential

- W$_{bsp}$: Weight assigned to the contribution of Budget Space Potential to the country's overall vulnerability score.

- CTB: Level of exposure from a country's Capacity to Borrow

- W$_{ctb}$: Weight assigned to the contribution of the to the country's capacity to borrow to their overall vulnerability score.

## Grouping the countries by vulnerability category

To enhance targeted support to countries, country vulnerability grouping/archetypes was explored. This was informed by the distribution and clustering of the overall vulnerability scores. Countries with scores below 40% were classified as having low vulnerability, those scoring between 40% and 59.5% were deemed moderately vulnerable, scores from 59.5% to 73% indicated high vulnerability, and scores of more than 73% were categorized as very high vulnerability. These were based on the empirical distribution of vulnerability scores (range 22.6–91.6; mean ≈ 60; standard deviation, SD ≈ 18) which revealed those four natural clusters. Scores ≥ 75% lie about one SD above the mean, defining the upper tail of extreme vulnerability. The 60–73% range approximates values around the mean ± 0.5 SD, representing moderately elevated vulnerability. Scores between 40–59.5% fall roughly one SD below the mean, indicating moderate vulnerability, while scores < 40% occupy the lower quartile.

## Validity assessment

To assess the validity of the derived vulnerability score, we explored the extent to which it explained cross-country differences in universal health coverage (UHC), as measured by the service coverage index (SCI); a composite index derived from 14 core indicators [39]. The choice of this measure was informed by overwhelming evidence that the level of health spending positively correlates with attainment of UHC SCI and other health outcome measures [40–42]. Given that the current scores gauge vulnerability of health financing systems, we anticipated that if the measure is to be externally valid, a negative correlation will be found against the UHC SCI. We therefore performed a linear regression between the vulnerability score and UHC SCI. Additionally, using latest available data on financial risk protection from 41 countries, we tested the correlation between the vulnerability scores and the number of people pushed further into poverty due to out-of-pocket health spending.

## Sensitivity analysis

Two types of uncertainty analysis are often considered in modelling: structural uncertainty analysis and parameter uncertainty analysis [43,44]. On the structural uncertainty analysis, one could argue, if domestic government spending is constrained and external aid is reduced or cut, the population may have to pay out-of-pocket for services that would otherwise

have been provided. However, the population might also forego healthcare entirely if the services are deemed unaffordable relative to their level of income [45–48]. Individual's decision to either increase out-of-pocket spending or forego services depends heavily on their socioeconomic status [45], particularly the level of poverty faced by the population and how it might crowd out other critical non-health spending [49,50]. Therefore, we included multi-dimensional poverty prevalence as a component in the analytical framework to explore its impact on vulnerability scores for countries.

On parameter uncertainty, the variables used lacked plausible ranges, as they were reported by placing countries into categories or as point expenditure estimates for countries without boundaries of uncertainty. Consequently, it was not feasible to conduct probabilistic sensitivity analysis. Consistent with best practice [44,51,52], we examined the impact of using different weighting methods on the vulnerability scores, considering exploratory factor analysis, entropy-based weighting, and equal weighting of all variables.

## 3. Results

### Descriptive summary of the health financing and macro-fiscal factors of exposure

**Current health expenditure per capita.** As shown in Fig 3, the latest available data show that in 2022, the current health expenditure per capita in the WHO African Region exhibited a seven-fold variation between countries, ranging from US$16.32 in Burundi to US$726.9 in Seychelles, averaging US$132.6 per capita. Over the decade between 2012 and 2022, the average current health expenditure per capita is about US$125.2 but fluctuates from one year to another (Fig 4) following

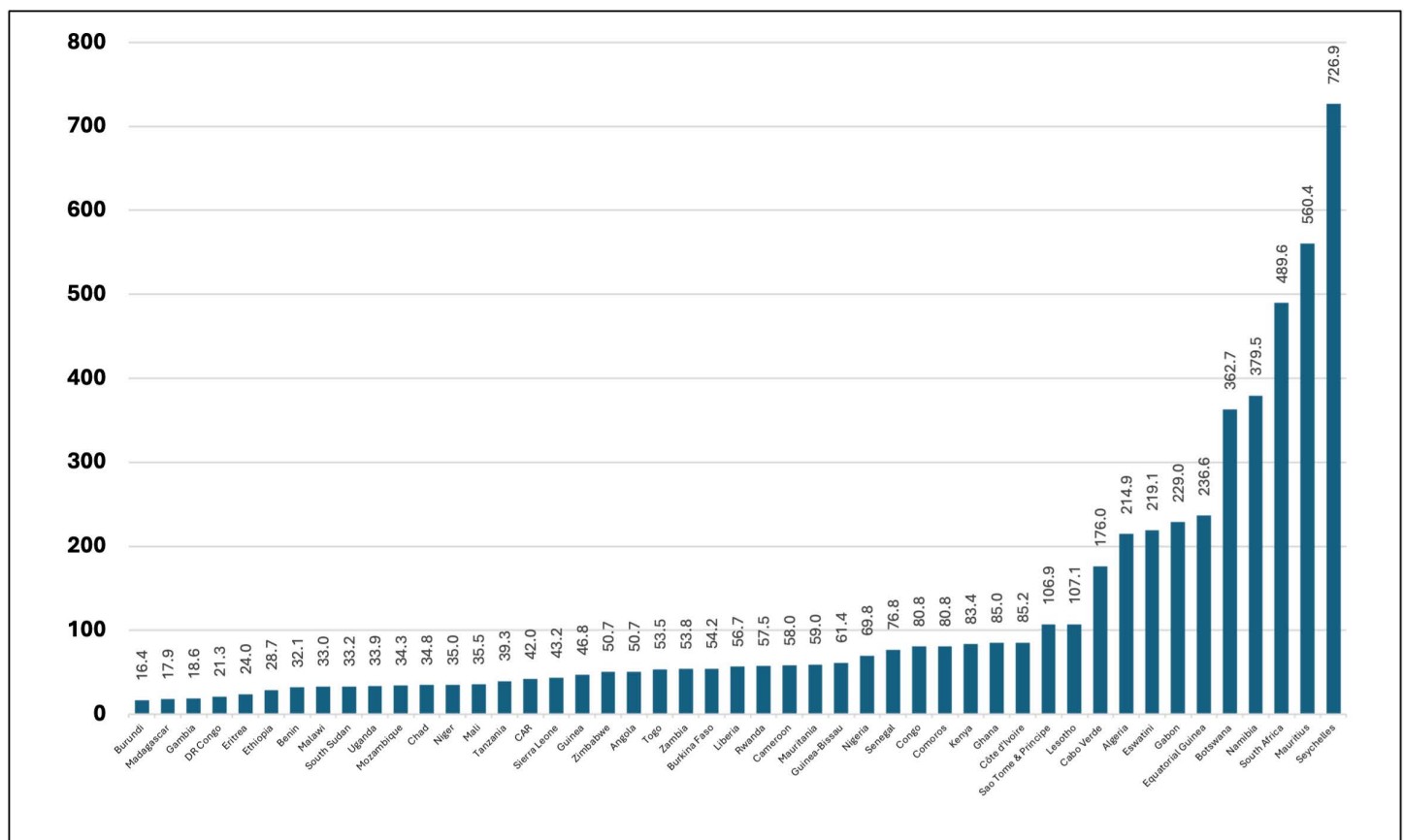

**Fig 3. Current Health Expenditure Per Capita in 2022 among countries in the WHO African Region.**

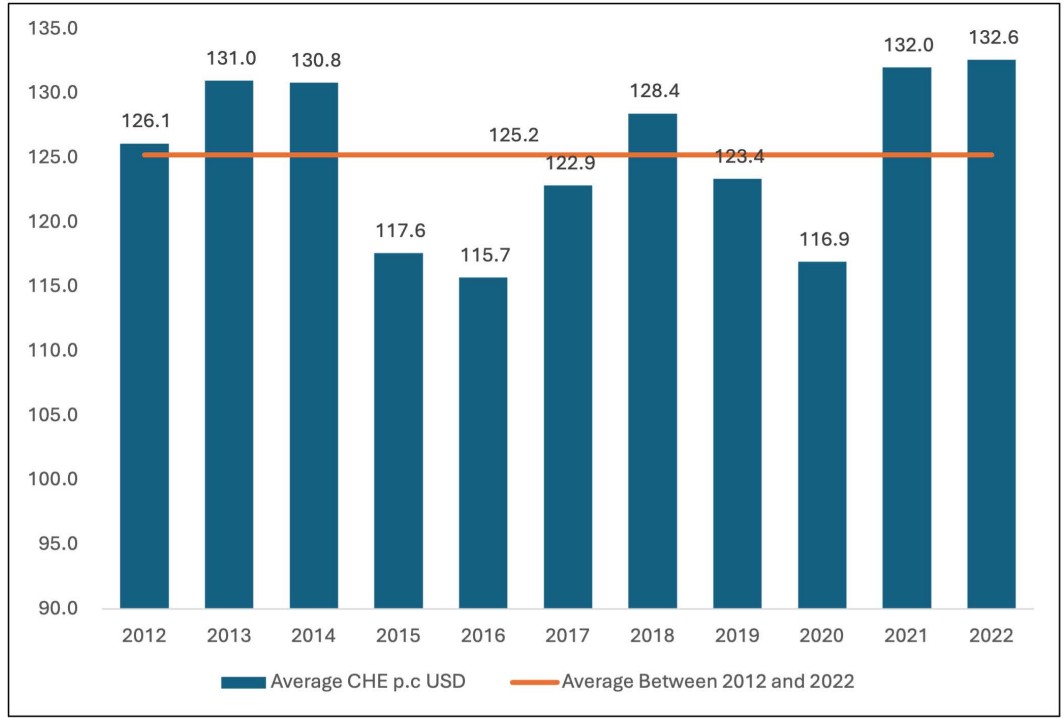

**Fig 4. Trends in the Average Current Health Expenditure Per Capita in the WHO African Region.**

a cycle of improvement and depression every 2–3 years. In the series, 2015, 2016 and 2020 stand out as periods where the regional average health spending per capita fell below US$120 per capita.

**Foreign/external aid dependency (current health expenditure from external sources).** As demonstrated in Fig 5, in 2022, external health expenditures constituted 23.9% of current health expenditures and have remained largely unchanged for a decade, from 23.4% in 2012. The private sector's contribution to health spending in the African region has been between 7.6% in 2012 and 8.5% in 2022. Domestic government spending and out-of-pocket spending on health were at almost similar proportions in 2022, respectively with 34.3% and 34.5% of total health spending. The proportion of health spending covered by domestic government has on average recorded a marginal shift over the last decade from 31.9% in 2012 to 34.2% in 2022.

At country level there are important differences in the level of health expenditures coming from external sources (Fig 6). In at least 5 countries (10.6% of the region), external funding contributes 45% or more (range: 45% to 65%) to their health expenditure, and in 17% of countries (n = 8), external funding contributes between 35% and 44% to their health expenditure, while in 6 countries (12.7%), external funding contributes between 25% and 34% to their health expenditure and in 36% of the countries (n = 17), external funding contributes between 15% and 24% to their health expenditure. Only eight countries (17% of the region) have external funding, contributing less than 15% to their health expenditure and relying more on domestic funding for their health systems, but external support still plays a role in their overall health financing.

**Budget space potential.** Using data over the last decade for analysis and projections of real general government expenditure (GGE) per capita, the World Bank classified countries into three categories as either "expansion", "stagnation", or "contraction". According to the analysis, countries in the contraction category are projected to experience absolute declines in real GGE per capita between 2019 and 2029 while countries in the stagnation category is projected to experience sluggish growth in real GGE per capita over the decade from 2019 to 2029. Those in the expansion category

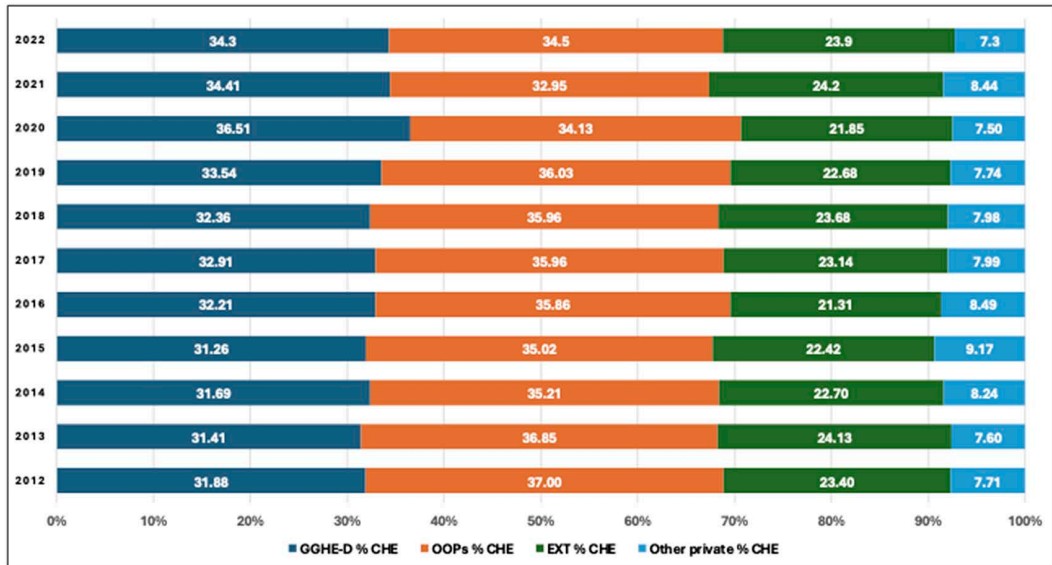

**Fig 5. Sources of Health Expenditure in the Africa Region 2012 - 2022.**

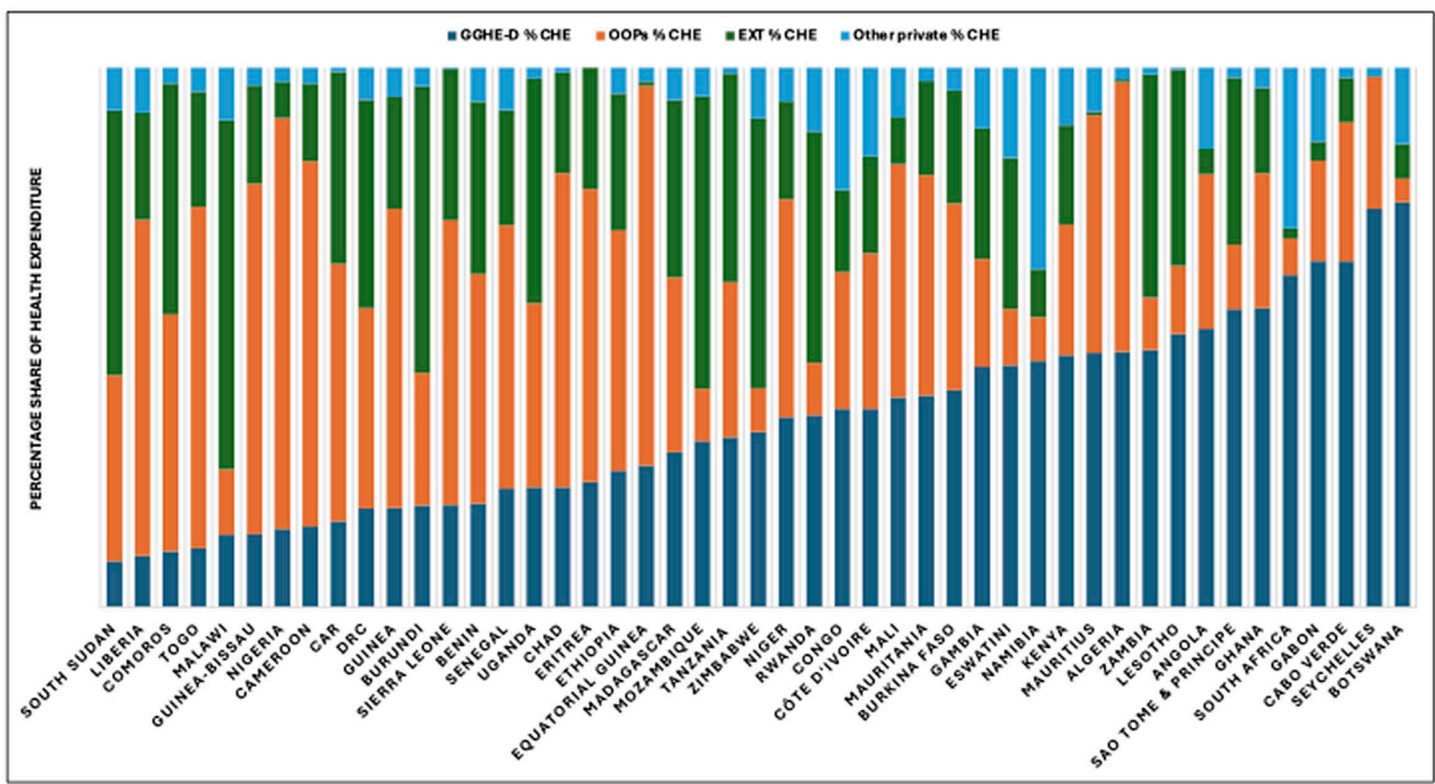

**Fig 6. Current Health Expenditure by Financing Sources, African Region, 2022.**

are projected to experience substantial growth in real GGE per capita between 2019 and 2029. Accordingly, the World Bank analysis suggests that almost 26% of countries in the African region (n = 12) are likely to experience a contraction in the general government expenditure, including spending in health. A higher proportion, almost 43%, are projected to record a stagnant general government expenditure through 2029, with only 32% of countries (n = 15) likely to expand general government expenditure, including health (see Table 2).

**Capacity to borrow.** A descriptive analysis of the latest debt sustainability assessment jointly published by the World Bank and IMF (summarised in Table 3) indicates that 13% of the countries in the WHO African region (n = 6) was already in debt distress by March 2025 while 17 countries (36%) were at high risk of debt distress. Only a quarter of the countries (25.5%, n = 12) were assessed to be facing moderate risk of debt distress. Another 25.5% of the countries were not assessed or included in the IMF/World Bank Low Income Countries Debt Sustainability Assessment (LIC DSA) framework due to being in higher income classification, or not in the International Development Association (IDA) lending eligibility. Thus, they are mostly in better macroeconomic situations than those listed in the LIC DSA framework, hence we assumed low risk of debt distress for the purpose of this analysis.

### Estimated health financing vulnerability scores of countries

Using the proposed framework and methodology and based on the latest available data on each variable, the overall vulnerability of countries in the African region averages 59.5%. Thus, the average vulnerability of national health systems to external financing shocks, particularly in the context of aid cuts, freezes, or withdrawals appears to be moderate. However, this varies substantially between countries, with some exhibiting lower vulnerability with a minimum score of 22.6%, while others face significantly unfavourable outlooks, reaching a maximum score of 91.6%. Thus, countries with high scores are considered structurally vulnerable to reductions in external health financing and may face service disruption or reversal of health gains without proactive mitigation measures. Table 4 provides a country-by-country assessment across the four dimensions and the overall vulnerability scores.

**Table 2. Summary of Budget Space Potential of Countries in the WHO African Region.**

| World Bank's Descriptor of Budget Space Potential | No of Countries | % of Countries |
|---|---|---|
| Expansion | 15 | 31.9% |
| Stagnation | 20 | 42.6% |
| Contraction | 12 | 25.5% |

Data source: World Bank (2024):

https://documents1.worldbank.org/curated/en/099110524145099363/pdf/P506692116ebcb0e188b4175eb-4c560cb5.pdf.

**Table 3. Summary of the Debt Sustainability of Countries in the WHO African Region.**

| World Bank (2024) Descriptor | No of Countries | % of Countries |
|---|---|---|
| In Debt Distress | 6 | 12.8% |
| High | 17 | 36.2% |
| Moderate | 12 | 25.5% |
| N/A | 12 | 25.5% |

**Data source**: https://www.worldbank.org/en/programs/debt-toolkit/dsa. N/A are countries not included in the IMF/World Bank LIC DSA framework due to being in higher income classification, or not in IDA lending eligibility.

**Table 4. Vulnerability Scores by Country.**

| Country | Scores Across the Variables | | | | Overall Vulnerability Score (max = 100%) | Vulnerability Grouping (Archetype) |
|---|---|---|---|---|---|---|
| | Weighted Level of Exposure from CHE per capita (max = 32.1%) | Weighted Level of Exposure from Foreign Aid Dependency (max = 30.7%) | Weighted Level of Exposure from Budget Space Potential (max = 7.6%) | Weighted Level of Exposure from Capacity to Borrow (max = 29.6%) | | |
| Malawi | 32.10 | 30.70 | 5.07 | 23.68 | 91.55 | Very High |
| Burundi | 32.10 | 30.70 | 7.60 | 17.76 | 88.16 | Very High |
| South Sudan | 32.10 | 30.70 | 7.60 | 17.76 | 88.16 | Very High |
| Mozambique | 32.10 | 30.70 | 5.07 | 17.76 | 85.63 | Very High |
| Ethiopia | 32.10 | 24.56 | 5.07 | 23.68 | 85.41 | Very High |
| Zimbabwe | 24.08 | 30.70 | 5.07 | 23.68 | 83.52 | Very High |
| Congo | 32.10 | 18.42 | 7.60 | 23.68 | 81.80 | Very High |
| Central African Republic | 32.10 | 24.56 | 5.07 | 17.76 | 79.49 | Very High |
| Sierra Leone | 32.10 | 24.56 | 5.07 | 17.76 | 79.49 | Very High |
| Zambia | 24.08 | 24.56 | 5.07 | 23.68 | 77.38 | Very High |
| Madagascar | 32.10 | 24.56 | 5.07 | 11.84 | 73.57 | Very High |
| Eritrea | 32.10 | 18.42 | 5.07 | 17.76 | 73.35 | Very High |
| Gambia | 32.10 | 18.42 | 5.07 | 17.76 | 73.35 | Very High |
| Niger | 32.10 | 18.42 | 5.07 | 17.76 | 73.35 | Very High |
| Benin | 32.10 | 24.56 | 2.53 | 11.84 | 71.03 | High. |
| Democratic Republic of the Congo | 32.10 | 24.56 | 2.53 | 11.84 | 71.03 | High. |
| Uganda | 32.10 | 24.56 | 2.53 | 11.84 | 71.03 | High. |
| Chad | 32.10 | 18.42 | 2.53 | 17.76 | 70.81 | High. |
| Togo | 32.10 | 18.42 | 2.53 | 17.76 | 70.81 | High. |
| Comoros | 16.05 | 24.56 | 7.60 | 17.76 | 65.97 | High. |
| Ghana | 24.08 | 18.42 | 5.07 | 17.76 | 65.32 | High. |
| Guinea-Bissau | 24.08 | 18.42 | 5.07 | 17.76 | 65.32 | High. |
| United Republic of Tanzania | 32.10 | 24.56 | 2.53 | 5.92 | 65.11 | High. |
| Rwanda | 24.08 | 24.56 | 2.53 | 11.84 | 63.01 | High. |
| Sao Tome and Principe | 8.03 | 24.56 | 5.07 | 23.68 | 61.33 | High. |
| Lesotho | 16.05 | 24.56 | 7.60 | 11.84 | 60.05 | High. |
| Liberia | 16.05 | 18.42 | 7.60 | 17.76 | 59.83 | High. |
| Cameroon | 24.08 | 12.28 | 5.07 | 17.76 | 59.18 | Moderate |
| Kenya | 16.05 | 18.42 | 5.07 | 17.76 | 57.30 | Moderate |
| Burkina Faso | 24.08 | 18.42 | 2.53 | 11.84 | 56.87 | Moderate |
| Guinea | 24.08 | 18.42 | 2.53 | 11.84 | 56.87 | Moderate |
| Senegal | 24.08 | 18.42 | 2.53 | 11.84 | 56.87 | Moderate |
| Mali | 32.10 | 6.14 | 5.07 | 11.84 | 55.15 | Moderate |
| Côte d'Ivoire | 16.05 | 18.42 | 2.53 | 11.84 | 48.84 | Moderate |
| Mauritania | 16.05 | 18.42 | 2.53 | 11.84 | 48.84 | Moderate |
| Eswatini | 8.03 | 24.56 | 5.07 | 5.92 | 43.57 | Moderate |
| Cabo Verde | 8.03 | 6.14 | 5.07 | 17.76 | 36.99 | Low |
| Algeria | 16.05 | 6.14 | 7.60 | 5.92 | 35.71 | Low |

*(Continued)*

**Table 4.** (Continued)

| Country | Scores Across the Variables | | | | Overall Vulnerability Score (max = 100%) | Vulnerability Grouping (Archetype) |
|---|---|---|---|---|---|---|
| | Weighted Level of Exposure from CHE per capita (max = 32.1%) | Weighted Level of Exposure from Foreign Aid Dependency (max = 30.7%) | Weighted Level of Exposure from Budget Space Potential (max = 7.6%) | Weighted Level of Exposure from Capacity to Borrow (max = 29.6%) | | |
| **Malawi** | **32.10** | **30.70** | **5.07** | **23.68** | **91.55** | **Very High** |
| Angola | 16.05 | 6.14 | 7.60 | 5.92 | 35.71 | Low |
| Nigeria | 16.05 | 6.14 | 2.53 | 5.92 | 30.64 | Low |
| Botswana | 8.03 | 6.14 | 7.60 | 5.92 | 27.69 | Low |
| Equatorial Guinea | 8.03 | 6.14 | 7.60 | 5.92 | 27.69 | Low |
| Namibia | 8.03 | 6.14 | 7.60 | 5.92 | 27.69 | Low |
| South Africa | 8.03 | 6.14 | 7.60 | 5.92 | 27.69 | Low |
| Mauritius | 8.03 | 6.14 | 5.07 | 5.92 | 25.15 | Low |
| Gabon | 8.03 | 6.14 | 2.53 | 5.92 | 22.62 | Low |
| Seychelles | 8.03 | 6.14 | 2.53 | 5.92 | 22.62 | Low |
| Average | **22.37** | **18.29** | **4.90** | **13.98** | **59.54** | |
| Minimum | **8.03** | **6.14** | **2.53** | **5.92** | **22.62** | |
| Maximum | **32.10** | **30.70** | **7.60** | **23.68** | **91.55** | |

Note: Countries with high scores are considered structurally vulnerable to reductions in external health financing and may face service disruption or reversal of health gains without proactive mitigation measures.

### Spatial distribution of the vulnerability archetypes

As illustrated in Fig 7 the spatial distribution of vulnerability categories among countries reveals a clustering of high or very high vulnerability in central, eastern, and southern African sub-regions. Conversely, West African countries predominantly exhibit moderate vulnerability, with a few falling into the very high and high categories.

### Validity analyses

**Relationship between the health financing vulnerability score and universal health converge.** In general, UHC SCI scores tended to decline as vulnerability scores increased (r=-0.56, p<0.001) (Fig 8), and a linear regression analysis further confirmed a moderate but statistically significant relationship, with vulnerability scores explaining 31.4% of the variation in UHC SCI across countries ($R^2$=0.314, p<0.001). Thus, increasing vulnerability in health system financing due to external aid freezes or cuts poses a significant risk of stalling or even reversing progress toward universal health coverage.

**Relationship between vulnerability and people impoverished by out-of-pocket payments.** Using latest available data on financial risk protection from 41 countries, the countries' vulnerability score had a positive and statistically significant correlation with the number of people pushed further into poverty due to out-of-pocket health spending (r=0.573, p<0.001). In a linear regression, the vulnerability scores explained about 32.8% of the variation in number of people pushed further into poverty due to out-of-pocket health spending across the countries ($R^2$=0.328, p<0.001). See **Fig 9** for a scatterplot of the relationship described.

### Sensitivity analysis

As demonstrated in Table 5, pre-existing poverty levels increased the average vulnerability score by 2.81% and altered the scores of 13 countries, 28% of the WHO African region. This led to 10 countries recording relatively high vulnerability scores, while 3 countries saw reduced scores. Nevertheless, as indicated in Table 6, the explanatory power of the model,

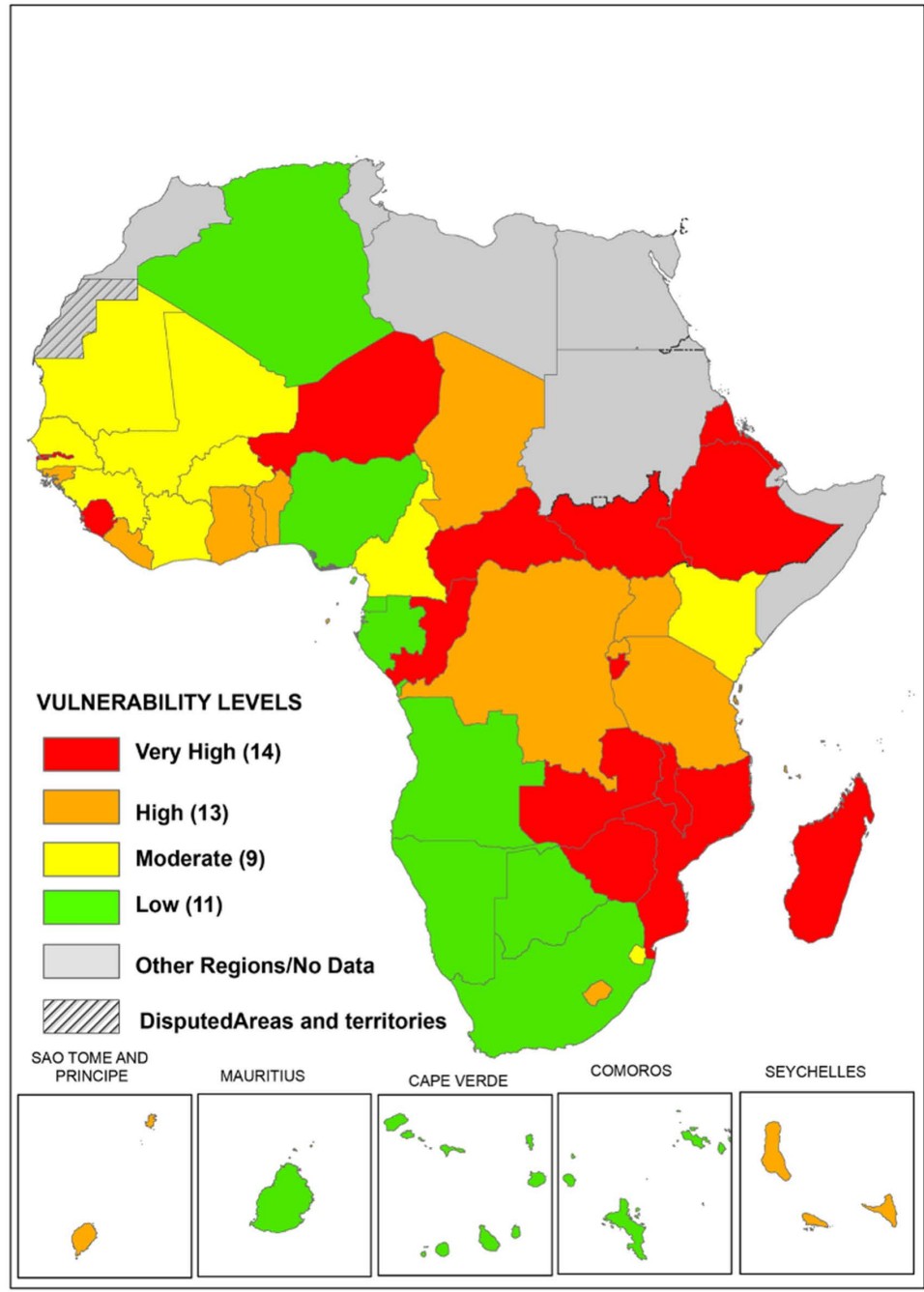

**Fig 7. Map visualizing vulnerability scores of countries in the WHO African Region. Data source** *Analysis by Health Systems and Services Cluster, WHO Regional Office for Africa Map production: GIS Team, WHO Regional Office for Africa, World Health Organization.*

a primary proxy for validity, decreased from 31% in our base model to 28% in the model with poverty included. As such, we favoured the parsimonious model with relatively higher explanatory power.

As outlined in Table 7, different methods of weighting variables produced vulnerability scores with relatively lower explanatory power compared to the base model results. Nevertheless, each approach resulted in scores that were highly

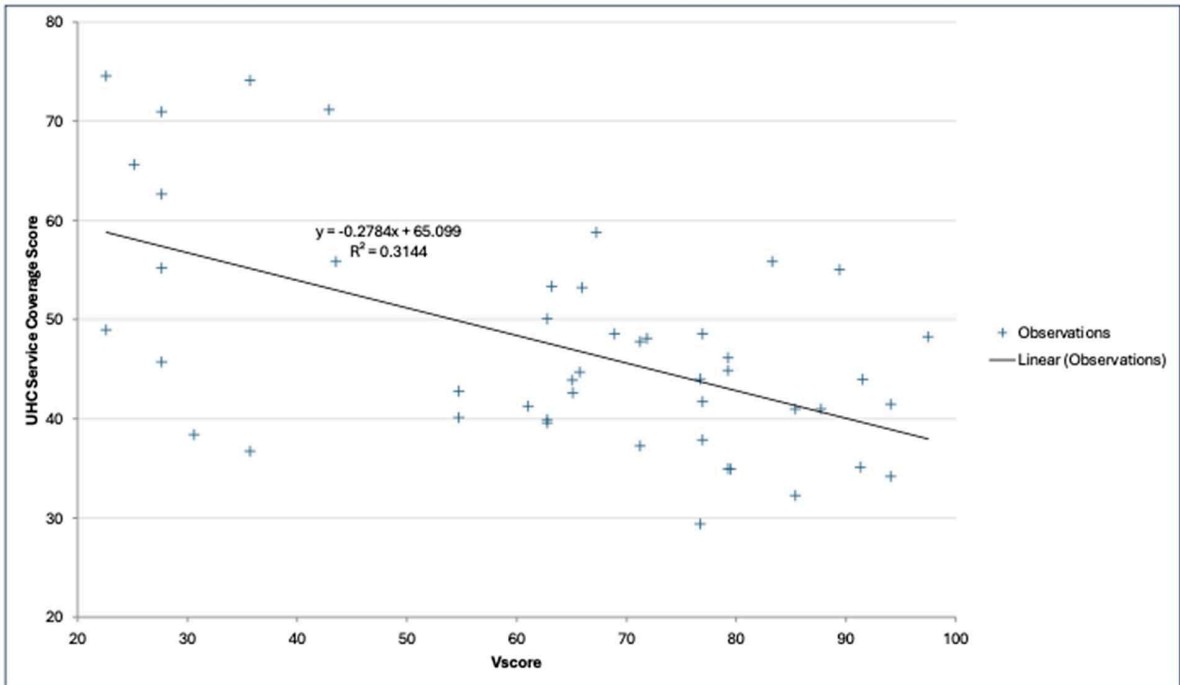

**Fig 8. Relationship between Vulnerability Score & UHC Service Coverage Index (UHC SCI).**

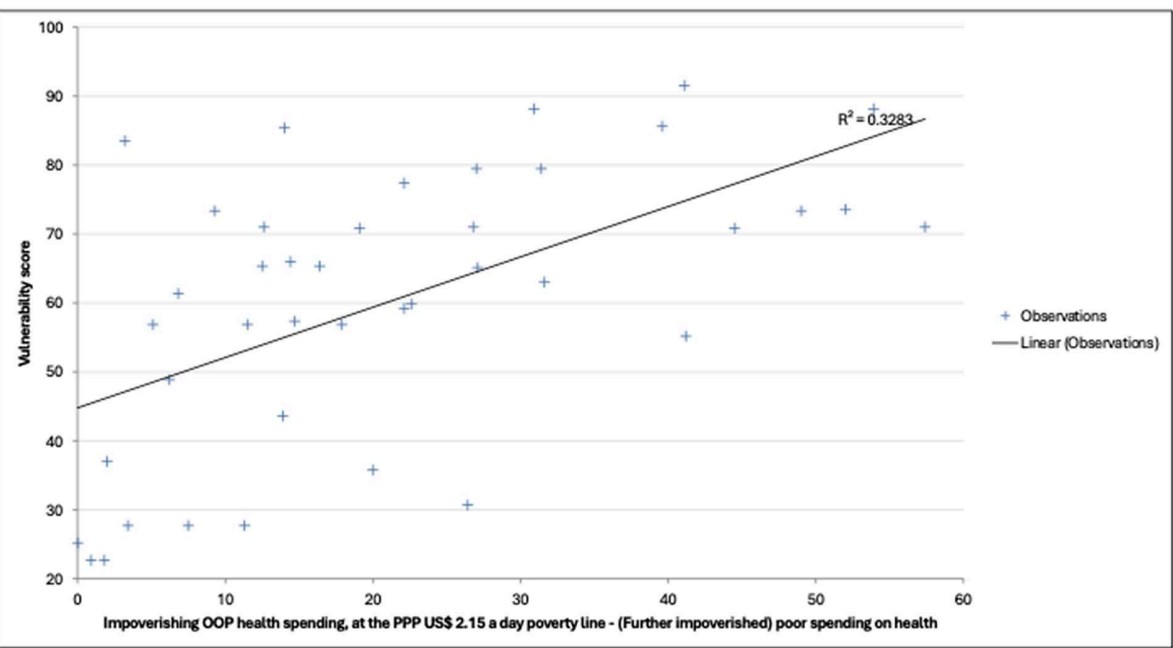

**Fig 9. How Vulnerability Scores Relate to Impoverishing Health Spending at the $2.15/Day Poverty Line.**

PLOS Global Public Health

**Table 5. Vulnerability Scores of Countries based on different methods of weighting variables and by inclusion of poverty as a variable.**

| Country | Overall Vulnerability Score (Polychoric Principal Components Analysis) | Overall Vulnerability Score (Exploratory Factor Analysis) | Overall Vulnerability Score (Entropy Method) | Overall Vulnerability Score (Unweighted) | Overall Vulnerability Score, structural change by including Poverty as a variable (Unweighted). |
|---|---|---|---|---|---|
| Algeria | 35.71 | 38.06 | 42.88 | 47.50 | 48.00 |
| Angola | 35.71 | 38.06 | 42.88 | 47.50 | 48.00 |
| Benin | 71.03 | 68.31 | 64.75 | 63.33 | 64.67 |
| Botswana | 27.69 | 30.76 | 36.68 | 41.25 | 38.00 |
| Burkina Faso | 56.87 | 55.25 | 52.97 | 52.08 | 60.67 |
| Burundi | 88.16 | 87.90 | 88.80 | 90.00 | 96.00 |
| Cabo Verde | 36.99 | 39.09 | 41.45 | 42.92 | 43.33 |
| Cameroon | 59.18 | 59.45 | 59.43 | 60.42 | 62.33 |
| Central African Republic | 79.49 | 78.27 | 76.79 | 76.67 | 75.33 |
| Chad | 70.81 | 68.65 | 64.77 | 63.33 | 74.67 |
| Comoros | 65.97 | 67.54 | 70.82 | 72.50 | 67.00 |
| Congo | 81.80 | 82.48 | 83.24 | 85.00 | 82.00 |
| Côte d'Ivoire | 48.84 | 47.95 | 46.77 | 45.83 | 45.67 |
| Democratic Republic of the Congo | 71.03 | 68.31 | 64.75 | 63.33 | 74.67 |
| Equatorial Guinea | 27.69 | 30.76 | 36.68 | 41.25 | 43.00 |
| Eritrea | 73.35 | 72.51 | 71.21 | 71.67 | 71.33 |
| Eswatini | 43.57 | 44.17 | 46.99 | 47.92 | 48.33 |
| Ethiopia | 85.41 | 84.37 | 82.39 | 81.67 | 89.33 |
| Gabon | 22.62 | 23.03 | 23.81 | 24.58 | 24.67 |
| Gambia | 73.35 | 72.51 | 71.21 | 71.67 | 66.33 |
| Ghana | 65.32 | 65.21 | 65.01 | 65.42 | 61.33 |
| Guinea | 56.87 | 55.25 | 52.97 | 52.08 | 60.67 |
| Guinea-Bissau | 65.32 | 65.21 | 65.01 | 65.42 | 61.33 |
| Kenya | 57.30 | 57.91 | 58.81 | 59.17 | 56.33 |
| Lesotho | 60.05 | 61.44 | 65.22 | 67.50 | 63.00 |
| Liberia | 59.83 | 61.78 | 65.24 | 67.50 | 73.00 |
| Madagascar | 73.57 | 72.17 | 71.19 | 71.67 | 81.33 |
| Malawi | 91.55 | 90.13 | 87.97 | 86.67 | 93.33 |
| Mali | 55.15 | 54.89 | 54.45 | 56.67 | 59.33 |
| Mauritania | 48.84 | 47.95 | 46.77 | 45.83 | 50.67 |
| Mauritius | 25.15 | 26.89 | 30.25 | 32.92 | 36.33 |
| Mozambique | 85.63 | 84.03 | 82.37 | 81.67 | 89.33 |
| Namibia | 27.69 | 30.76 | 36.68 | 41.25 | 38.00 |
| Niger | 73.35 | 72.51 | 71.21 | 71.67 | 81.33 |
| Nigeria | 30.64 | 30.33 | 30.01 | 30.83 | 29.67 |
| Rwanda | 63.01 | 61.01 | 58.55 | 57.08 | 64.67 |
| Sao Tome and Principe | 61.33 | 62.47 | 63.79 | 62.92 | 59.33 |
| Senegal | 56.87 | 55.25 | 52.97 | 52.08 | 50.67 |
| Seychelles | 22.62 | 23.03 | 23.81 | 24.58 | 29.67 |
| Sierra Leone | 79.49 | 78.27 | 76.79 | 76.67 | 80.33 |
| South Africa | 27.69 | 30.76 | 36.68 | 41.25 | 38.00 |
| South Sudan | 88.16 | 87.90 | 88.80 | 90.00 | 96.00 |

*(Continued)*

**Table 5.** (Continued)

| Country | Overall Vulnerability Score (Polychoric Principal Components Analysis) | Overall Vulnerability Score (Exploratory Factor Analysis) | Overall Vulnerability Score (Entropy Method) | Overall Vulnerability Score (Unweighted) | Overall Vulnerability Score, structural change by including Poverty as a variable (Unweighted). |
|---|---|---|---|---|---|
| Togo | 70.81 | 68.65 | 64.77 | 63.33 | 64.67 |
| Uganda | 71.03 | 68.31 | 64.75 | 63.33 | 69.67 |
| United Republic of Tanzania | 65.11 | 62.21 | 59.15 | 58.33 | 61.67 |
| Zambia | 77.38 | 77.07 | 76.19 | 75.42 | 79.33 |
| Zimbabwe | 83.52 | 82.83 | 81.77 | 80.42 | 78.33 |
| **Average** | **59.54** | **59.40** | **59.58** | **60.26** | **62.35** |
| **Minimum** | **22.62** | **23.03** | **23.81** | **24.58** | **24.67** |
| **Maximum** | **91.55** | **90.13** | **88.80** | **90.00** | **96.00** |

**Table 6. Alternative Methods and their Relationship with UHC SCI.**

| Alternative Methods and their relationship with UHC SCI | R | R² |
|---|---|---|
| Overall Vulnerability Score (Polychoric Principal Components Analysis) | -0.56 | 0.31 |
| Overall Vulnerability Score (Exploratory Factor Analysis) | -0.54 | 0.29 |
| Overall Vulnerability Score (Entropy Method) | -0.50 | 0.25 |
| Overall Vulnerability Score (Unweighted) | -0.47 | 0.23 |
| Overall Vulnerability Score, structural change by including Poverty as a variable (Unweighted) | -0.53 | 0.28 |

**Table 7. Relationship between V-score of Polychroric PCA and V-scores of Other Methods.**

| Relationship V-score of Polychroric PCA and V-scores of Other Methods | R | R² |
|---|---|---|
| Polychoric PCA vs EFA | 0.998 | 0.995 |
| Polychoric PCA vs Entropy | 0.980 | 0.960 |
| Polychoric PCA vs Unweighted | 0.957 | 0.916 |
| Polychoric PCA vs Poverty Unweighted | 0.959 | 0.920 |

correlated with those of the base model. As a result, any of these methods could, in principle, be fit-for-purpose when analysing countries or informing decision-making. However, we maintained the base model results derived from applying the weights of polychoric PCA, as it demonstrated the highest explanatory power, albeit marginally.

## 4. Discussion

The health sector has been the number one priority area for external aid coming into Africa, receiving 23% of the over US$16 billion annual aid disbursement from the countries of OECD, hence external aid cuts are expected to adversely impact the financing of health systems in Africa. Using our analytical framework to conduct a desk-based assessment, the overall vulnerability score of countries in the African region averages almost 60%, varying substantially from 23% in countries exhibiting lowest vulnerability, such as Seychelles to 92% in countries with the highest vulnerability, such as Malawi.

Spatial analysis shows a clustering of high levels of vulnerability in East and Southern African countries due to a combination of high levels of donor dependence and unfavourable macro-fiscal outlook. Our analysis further reveals that all countries do not face the same level of risk or have varying levels of resilience to these financial shocks. The adverse impact of sudden cuts or freezes in external aid for health has been dire for population health, significantly heightening the vulnerabilities of health systems. The vulnerability assessed in this paper is negatively associated with progress on

universal health coverage, explaining 31% (p<0.001) – hence could potentially precipitate reversal in the progress made on universal health coverage if nothing is done to contain or mitigate. To put this in context, a recent modelling study in South Africa revealed that reduced PEPFAR funding without appropriate mitigation measures could increase new HIV infections between 286,000 and 565,000 over 10 years (24% - 47.4% increase) and reverse life expectancy by 2–4 years for people living with HIV [53].

To the best of our knowledge, this paper is the first attempt to quantify the vulnerability of health financing occasioned by external aid cuts, particularly in the African context where external aid has been an important part of health investments. There is, therefore, no findings of prior assessments of this nature to compare our findings to. However, the result emerging from the assessment is intuitive and of practical significance, helping to identify countries at most risk even if they are not necessarily the biggest recipients of external aid in volume. Box 1 provides contextual examples using Nigeria and Malawi. For instance, Malawi ranks as the 10th largest recipient of health-related external aid by volume in the African region but is ranked highest in terms of its vulnerability from our assessment. In contrast, Nigeria and South Africa which are the top two recipients of United States' external aid by volume in the WHO African region has been shown to rather have relatively lower vulnerabilities of 30.64% and 27.69%, respectively – thus, having an inherent potential to mitigate the impact of the aid transition, at least in the short term. Consistent with this assessment outcome, South Africa's parliament has reportedly approved an additional US$1.5 billion to augment its health budget, mainly to cover salaries of some 9,300 health workers in clinics and hospitals and about 800 newly qualified doctors [16,54]. Similarly, Nigeria has allocated an extra US$200 million to mitigate the effects of aid cuts on service delivery, while Kenya's health ministry has requested an additional US$250 million from the National Treasury [55].

> ## Box 1. Case study of Malawi and Nigeria.
>
> Malawi has a relatively unfavourable outlook across the four domains, with CHE per capita of only ~US$ 40 (in the 20th percentile) and 64.7% of this spending financed from external aid. World Bank projections suggest that general government expenditure, including health, would likely remain stagnant up to 2029, while the country is classified by the World Bank and IMF as facing debt distress (hence with limited options to borrow for health). These factors produce a composite vulnerability score of 92%, placing Malawi among the countries with "very high vulnerability" if there are sudden or even progressive external aid freezes/cuts in the short-to-medium term. A 40% reduction in United States health assistance would withdraw an estimated US$ 106 million; a complete suspension could result in US$ 177 million in forgone health investments, equivalent to 14–23% of Malawi's projected 2025 CHE–well beyond what domestic resources could plausibly replace in a dollar-to-dollar manner in the short-to-medium term.
>
> Compared with Malawi, Nigeria receives larger absolute volumes of donor support yet exhibits a markedly lower vulnerability score of 31%. Its CHE per capita is ~US$ 91 (in the 60th percentile), with external aid accounting for just 6.8% of Nigeria's current health spending. The World Bank projections suggest that the country has an expansionary potential for general government expenditure, including domestic public health spending, and the country is not classified as in debt distress by the World Bank and IMF, affording room to borrow for health if required.

Some variables such as patients paying out of pocket will be one of the means of coping in the context of aid cuts. However, this will also depend on the level of poverty in the country (or ability to spend more among the population). Despite its conceptual relevance, we observe that slight decrease in explanatory power ($R^2$ dropped from 31% to 28%) when the poverty variable was included. This may not necessarily suggest that poverty was a less important variable but may require further assessment to understand the nature of its contribution. Several studies, particularly from China has demonstrated the role of poverty reduction in access to health services and mitigating households vulnerabilities in

relation to catastrophic out-of-pocket payments [20–22,56]. One key challenge we observed is that the poverty data for most countries are outdated (in some cases about 5–10 years old) as it is derived from national surveys which are not always regular or undertaken the same year for all countries. As a result, temporality with the rest of the variables included in the analysis appear to be limited. Thus, the replicability and policy-tracking utility of the proposed measurement framework depends on the routine and timely updating of the underlying datasets for all constituent variables. The sensitivity observed with the poverty data in this analysis demonstrates that delays or irregular updates in any of these datasets can influence the responsiveness of the framework. Future work should thus, should not only focus on the missing piece about the role of poverty, but also the extent to which the external aid cuts which permeates all sectors will have on poverty levels. It will be important for future work to explore multisectoral vulnerability, beyond focusing on one sector, as explored in this paper, which could help pinpoint countries at risk of greater financial shocks which could push them into a more long-term fragile or vulnerable state.

Even though the full extent of the impact and the true level of vulnerability of countries is not yet clear and might take years to be ascertained, beyond assessing the level of vulnerability, there is a need to begin counting the lives that would likely be lost due to the aid cuts - a call already made by other eminent scientists [57]. However, as countries' data systems are also hard hit by the aid cuts, innovative methodological approaches combining inexpensive data collection and modelling are required in counting the losses in population health and health systems capacity and performance - and all hands should be on the deck.

Looking ahead, there could be some potential uses of the vulnerability score, such as its inclusion in considering readiness of a country to transition from support of the Global health Initiatives (GHIs) such as the Global Fund and GAVI, as well as its consideration in future analyses or discussions on strengthening health system resilience. It is also worth noting that the vulnerability of countries to external aid reductions will not be static and could evolve significantly based on national responses and policy adjustments. Countries that proactively implement mitigation strategies such as expanding domestic resource mobilization, improving health budget execution, diversifying funding sources, and strengthening public financial management could reduce their exposure to external shocks over time. For instance, increased reliance on progressive taxation or health insurance schemes could reduce foreign aid dependency, while improved fiscal discipline and debt restructuring may enhance borrowing capacity. Additionally, strategic investments in data systems and monitoring frameworks would allow for early detection of financing gaps and facilitate timely corrective measures. Therefore, periodic reassessment of vulnerability scores will be essential to reflect policy changes, as part of the regular Africa Health Financing atlas produced by WHO.

## 5. Limitations of the study

### Conceptual issues

Even though the conceptual framework has helped to effectively identify key factors influencing health financing vulnerability (namely: CHE per capita, dependency on foreign aid, potential budget space and capacity to borrow), it does not address all options of potentially influential variables. We, therefore, acknowledge that the framework and its resulting vulnerability scores should be interpreted as a rough guide as some key variables that were not included could be considered in further theorizing the framework as more evidence on their role emerges. For example, variables such as poverty levels and private sector contribution to offset deprioritized services from government were not considered. Also, further borrowing capacity may be politically constrained beyond technical debt sustainability. The inclusion of these variables and others would require further investigation to further enhance the proposed approach.

### Depth of analysis

It is also worth noting that this assessment primarily focused on the health system level and did not attempt to breakdown by disease-specific programs. The available data indicates strongly that more than 70% of all the health-related

aid coming to Africa focused on HIV/AIDS and reproductive, maternal and child health, hence those programs could be disproportionately affected by the ongoing aid cuts. Nevertheless, it is feasible to adapt the model developed in this paper for programmatic level analyses within a country or across countries.

### Geographical scope

The analysis was intended to inform WHO's policy guidance and response for countries in the WHO African region, which is made up of the 47 countries included in this paper. The other countries in the continent of Africa that have not been included belongs to the WHO Eastern Mediterranean Region and hence were out of the scope of this work. Nonetheless, the model developed is easily adaptable to assess the vulnerability of those or any other country if data are available.

### Data limitations

The study relied on data from publicly available sources from the World Health Organization, the World Bank, and the International Monetary Fund, hence, the inherent limitations in the estimations from those sources are indirectly inherited in this study. While the study relied on the most recent data available from standardized sources, some countries report actual expenditures with delays, while others rely on projections or modelled estimates. In addition, some of the latest debt sustainability assessments are more than 2 years old and may not always fully reflect the current fiscal realities of countries, especially those experiencing rapid economic changes. Further, we assumed that countries not assessed by the IMF/World Bank as "low risk" of debt distress. However, this may not always be the case does not preclude countries in this group having macroeconomic vulnerabilities, which could significantly impact the health financing vulnerabilities derived in the current analysis. Thus, for those countries, under- or over-estimation of the level of health financing vulnerability is a possibility.

## 6. Policy and practice recommendations

To avoid unnecessary deaths and suffering and to make progress towards inclusive economic growth, countries need to protect essential health services by prioritizing the most vulnerable populations, protect their health budgets through increasing efficiency, generating new revenue and earmarking to health through taxes and other sources of funding from global partners, as well as better pooling of resources for health.

Countries facing debt distress and budget space contraction or stagnation will require pro-poor concessional financing for health while those with more favourable macro-fiscal outlook could also consider strategic borrowing for health. These debt-related financing mechanisms should target the most cost-effective investments, align with country priorities, and flow through the public financial management systems of the beneficiary countries to contribute to strengthening their macroeconomic foundations. Additionally, grants and debt-to-health swaps will be essential as well as private sector engagement to optimise their contribution.

In all countries, current and future external health financing should be underpinned by jointly developed sustainability plans, and external aid should be aligned with national priorities and flow through domestic budgets or national systems as much as possible. Finally, regional coordination, dialogues, and technical guidance are urgently needed. The WHO, together with its partners, is poised to support countries in exploring avenues for raising resources and providing technical advice to reconfigure their health systems. This support will help navigate the crisis while protecting progress towards universal health coverage and health security. S2 File provides a comprehensive list of recommendations for countries based on their level of vulnerability.

## Supporting information

**S1 Data. Raw dataset and analytics.**
(XLSX)

**S1 File. Vulnerability Assessment by Country.**
(DOCX)

**S2 File. Recommendations for countries based on vulnerability category.**
(DOCX)

## Acknowledgments

We acknowledge feedback and internal peer-review from team leaders across clusters in the WHO Regional Office for Africa, alongside the policy advisors and health financing technical Officers from the 47 WHO country offices in the African region.

## Author contributions

**Conceptualization:** James Avoka Asamani, Sophie Faye, Kouadjo San Boris Bediakon, Hilary Kipruto, Sunny C. Okoroafor, Janet Kayita, Benson Droti, Ogochukwu Chukwujekwu.

**Data curation:** James Avoka Asamani, Sophie Faye, Kouadjo San Boris Bediakon, Sunny C. Okoroafor, Ame Dioka, Azzam Ali, Benson Droti.

**Formal analysis:** James Avoka Asamani, Sophie Faye, Kouadjo San Boris Bediakon, Hilary Kipruto, Sunny C. Okoroafor, Ame Dioka, Azzam Ali, Benson Droti, Ogochukwu Chukwujekwu.

**Investigation:** James Avoka Asamani, Hilary Kipruto.

**Methodology:** James Avoka Asamani, Hilary Kipruto.

**Project administration:** Ame Dioka.

**Resources:** Hilary Kipruto.

**Software:** Kouadjo San Boris Bediakon.

**Supervision:** Ogochukwu Chukwujekwu.

**Validation:** James Avoka Asamani, Sophie Faye, Sunny C. Okoroafor, Janet Kayita, Benson Droti, Ogochukwu Chukwujekwu.

**Writing – original draft:** James Avoka Asamani, Benson Droti, Ogochukwu Chukwujekwu.

**Writing – review & editing:** James Avoka Asamani, Sophie Faye, Kouadjo San Boris Bediakon, Sunny C. Okoroafor, Janet Kayita, Ame Dioka, Azzam Ali, Ogochukwu Chukwujekwu.

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
