## [Decision Letter · Decision Letter 0]

29 Sep 2025

PGPH-D-25-02231

Measuring Health Financing Vulnerability Due to Reductions in Official Development Assistance: A Conceptual Framework with Empirical Application Across 47 African Countries

Dear Dr. Asamani,

Thank you for submitting your manuscript to PLOS Global Public Health. After careful consideration, we feel that it has merit but does not fully meet PLOS Global Public Health’s publication criteria as it currently stands. Therefore, we invite you to submit a revised version of the manuscript that addresses the points raised during the review process.

Your manuscript has been evaluated by one reviewer, and their comments are available below and in the attached document. Please carefully revise your manuscript to address the points raised.

Please note that we have only been able to secure a single reviewer to assess your manuscript. We are issuing a decision on your manuscript at this point to prevent further delays in the evaluation of your manuscript. Please be aware that the editor who handles your revised manuscript might find it necessary to invite additional reviewers to assess this work once the revised manuscript is submitted. However, we will aim to proceed on the basis of this single review if possible.

We look forward to receiving your revised manuscript.

Kind regards,

Jenna Scaramanga

Staff Editor

Journal Requirements:

Additional Editor Comments (if provided):

Reviewers' comments:

Reviewer's Responses to Questions

**Comments to the Author**

1. Does this manuscript meet PLOS Global Public Health’s publication criteria? Is the manuscript technically sound, and do the data support the conclusions? The manuscript must describe methodologically and ethically rigorous research with conclusions that are appropriately drawn based on the data presented.? Is the manuscript technically sound, and do the data support the conclusions? The manuscript must describe methodologically and ethically rigorous research with conclusions that are appropriately drawn based on the data presented.

Reviewer #1: Yes

2. Has the statistical analysis been performed appropriately and rigorously?

Reviewer #1: Yes

3. Have the authors made all data underlying the findings in their manuscript fully available (please refer to the Data Availability Statement at the start of the manuscript PDF file)?

The PLOS Data policy requires authors to make all data underlying the findings described in their manuscript fully available without restriction, with rare exception. The data should be provided as part of the manuscript or its supporting information, or deposited to a public repository. For example, in addition to summary statistics, the data points behind means, medians and variance measures should be available. If there are restrictions on publicly sharing data—e.g. participant privacy or use of data from a third party—those must be specified.requires authors to make all data underlying the findings described in their manuscript fully available without restriction, with rare exception. The data should be provided as part of the manuscript or its supporting information, or deposited to a public repository. For example, in addition to summary statistics, the data points behind means, medians and variance measures should be available. If there are restrictions on publicly sharing data—e.g. participant privacy or use of data from a third party—those must be specified.

Reviewer #1: Yes

4. Is the manuscript presented in an intelligible fashion and written in standard English?

Reviewer #1: Yes

5. Review Comments to the Author

Reviewer #1: Evaluation

This paper aims to construct a multidimensional framework to assess the financing vulnerability of health systems in African countries in the context of reductions in external assistance. Through an empirical analysis of 47 African countries, it provides evidence to inform policy-making. The research question is highly relevant and urgent; however, several aspects of the methodology and presentation require further refinement. It is recommended for publication after some revisions.

Comments

1. The Results section of the Abstract could briefly mention the spatial clustering of highly vulnerable countries, such as those in Eastern and Southern Africa.

2. As key background information, Figure 1 should be described in greater detail in the main text. In addition, briefly discussing the underlying reasons for aid cuts would deepen the contextual analysis.

3. The study adopts the economic vulnerability framework of Briguglio & Galea and Guillaumont. Please clarify why this framework was chosen over others, and consider briefly comparing it with existing vulnerability or resilience frameworks to highlight its relative advantages and suitability.

4. The analysis covers 47 WHO African Region countries. Please clarify why the other seven African countries were not included, and how their inclusion might have affected the overall findings and regional vulnerability patterns.

5. On page 6, when classifying countries into four exposure levels based on current health expenditure per capita, please explain why the 20th, 40th, 60th, and 80th percentiles were chosen. The division and cutoff values from the 60th to the 100th percentile require clarification.

6. Please check and correct minor editorial issues throughout the manuscript, such as the extra symbol on page 7, line breaks on pages 16 and 19, table numbering on pages 20-21, and the overall clarity of figures.

7. Why are the results of polychoric PCA presented in the Methodology section?

8. Regarding the “budget space potential” variable, please specify the criteria used by the World Bank to classify countries into “expansion”, “stagnation”, and “contraction”, and provide a corresponding reference.

9. Classifying countries not assessed by the IMF/World Bank (N/A) as “low risk” is a reasonable assumption, but this should be explicitly stated, and the potential implications (e.g., possible under- or over-estimation of risk for certain countries) should be discussed.

10. How were the thresholds for vulnerability categories determined (e.g., 40%, 59.5%, and 73% for “low”, “high”, and “very high” vulnerability)? Please provide a justification or cite relevant methodology.

11. In Figure 8, why do the coefficients of the two equations differ? Including two different equations in the same figure is puzzled.

12. The slight decrease in explanatory power (R2 dropped from 31% to 28%) when the poverty variable was included in Table 5 requires discussion.

13. The Discussion could benefit from a more in-depth case-style analysis of representative countries to illustrate why they fall into specific vulnerability categories.

14. Policy recommendations could be more specific and tailored to countries with “high”, “moderate”, and “low” levels of vulnerability, respectively.

15. The dynamic aspect of the framework could be further emphasized. It may be beneficial to periodically update the vulnerability scores to reflect policy changes, and a regular reassessment mechanism might be considered.

6. PLOS authors have the option to publish the peer review history of their article (what does this mean?). If published, this will include your full peer review and any attached files.). If published, this will include your full peer review and any attached files.

**Do you want your identity to be public for this peer review?** For information about this choice, including consent withdrawal, please see our Privacy Policy..

Reviewer #1: No

Figure Resubmissions:

---

## [Decision Letter · Decision Letter 1]

20 Nov 2025

PGPH-D-25-02231R1

Measuring Health Financing Vulnerability Due to Reductions in Official Development Assistance: A Conceptual Framework with Empirical Application Across 47 African Countries

Dear Dr. Asamani,

Thank you for submitting your manuscript to PLOS Global Public Health. After careful consideration, we feel that it has merit but does not fully meet PLOS Global Public Health’s publication criteria as it currently stands. Therefore, we invite you to submit a revised version of the manuscript that addresses the points raised during the review process.

We look forward to receiving your revised manuscript.

Kind regards,

Meless Gebrie Bore, PhD in Health

Academic Editor

Journal Requirements:

Additional Editor Comments (if provided):

Manuscript Title: Measuring Health Financing Vulnerability Due to Reductions in Official Development Assistance: A Conceptual Framework with Empirical Application Across 47 African Countries

Decision: Major Revision Required

Dear Authors,

Thank you for submitting your manuscript to our journal. The manuscript addresses an important and timely topic—assessing the financing vulnerability of health systems in African countries in the context of declining external assistance. The reviewer acknowledges the relevance and policy importance of your work and recommends the manuscript for publication after revisions.

After carefully considering the reviewer’s report, I request that you revise your manuscript to address the points outlined below. Please submit a revised manuscript, both in clean form and with tracked changes, along with a detailed point-by-point response.

Summary of Required Revisions

Based on the reviewer’s comments, please address the following key issues:

1. Abstract

• Include a brief description of the spatial clustering of highly vulnerable countries (e.g., Eastern and Southern Africa).

2. Background and Context

• Expand the explanation of Figure 1 in the main text.

• Provide brief contextual information on the underlying causes of recent reductions in external aid.

3. Methodological Clarifications

• Justify the selection of the Briguglio & Galea / Guillaumont economic vulnerability framework compared to other existing frameworks.

• Clarify why analysis was restricted to 47 WHO African Region countries and discuss the potential implications of excluding the remaining seven African countries.

• Explain the rationale for using the 20th, 40th, 60th, and 80th percentiles to classify exposure levels, and clarify cutoffs for the upper categories.

• Move presentation of polychoric PCA results from the Methods section to the Results section.

• Provide the World Bank criteria and citation for classification of “budget space potential” (expansion/stagnation/contraction).

• Explicitly justify the assumption that countries classified as N/A in IMF/World Bank assessment are “low risk,” and discuss its potential implications.

• Explain how thresholds for vulnerability categories (e.g., 40%, 59.5%, 73%) were determined.

4. Results Section

• Clarify why Figure 8 includes two equations with differing coefficients.

• Discuss the slight decline in R² in Table 5 when the poverty variable is added.

5. Discussion and Policy Implications

• Strengthen the discussion by including brief case-style insights into representative countries to illustrate vulnerability patterns.

• Provide more specific and differentiated policy recommendations targeted to countries with high, moderate, and low vulnerability.

• Highlight the dynamic nature of the proposed framework and discuss the importance of periodic reassessment.

6. Editorial and Formatting Corrections

• Correct minor editorial errors identified by the reviewer (e.g., stray symbols, inconsistent line breaks, table numbering, figure clarity).

Submission Requirements

Please upload the following:

1. Revised manuscript (clean version).

2. Revised manuscript (tracked changes).

3. Point-by-point response to each reviewer comment, indicating where changes were made (page and line numbers).

4. Updated figures and tables, if modified.

We look forward to receiving your revised manuscript and believe that the proposed improvements will significantly strengthen the clarity, methodological rigor, and policy relevance of your work.

Please do not hesitate to reach out if clarification is needed.

Sincerely,

Dr Meless Gebrie Bore

Associate Editor / Handling Editor

Reviewers' comments:

Reviewer's Responses to Questions

**Comments to the Author**

1. If the authors have adequately addressed your comments raised in a previous round of review and you feel that this manuscript is now acceptable for publication, you may indicate that here to bypass the “Comments to the Author” section, enter your conflict of interest statement in the “Confidential to Editor” section, and submit your "Accept" recommendation.

Reviewer #1: All comments have been addressed

2. Does this manuscript meet PLOS Global Public Health’s publication criteria? Is the manuscript technically sound, and do the data support the conclusions? The manuscript must describe methodologically and ethically rigorous research with conclusions that are appropriately drawn based on the data presented.? Is the manuscript technically sound, and do the data support the conclusions? The manuscript must describe methodologically and ethically rigorous research with conclusions that are appropriately drawn based on the data presented.

Reviewer #1: Yes

3. Has the statistical analysis been performed appropriately and rigorously?

Reviewer #1: Yes

4. Have the authors made all data underlying the findings in their manuscript fully available (please refer to the Data Availability Statement at the start of the manuscript PDF file)?

The PLOS Data policy requires authors to make all data underlying the findings described in their manuscript fully available without restriction, with rare exception. The data should be provided as part of the manuscript or its supporting information, or deposited to a public repository. For example, in addition to summary statistics, the data points behind means, medians and variance measures should be available. If there are restrictions on publicly sharing data—e.g. participant privacy or use of data from a third party—those must be specified.requires authors to make all data underlying the findings described in their manuscript fully available without restriction, with rare exception. The data should be provided as part of the manuscript or its supporting information, or deposited to a public repository. For example, in addition to summary statistics, the data points behind means, medians and variance measures should be available. If there are restrictions on publicly sharing data—e.g. participant privacy or use of data from a third party—those must be specified.

Reviewer #1: Yes

5. Is the manuscript presented in an intelligible fashion and written in standard English?

Reviewer #1: Yes

6. Review Comments to the Author

Reviewer #1: The authors have adequately addressed most of my previous concerns. The manuscript has been significantly improved through revisions that enhance its clarity, methodological rigor, and policy relevance.

However, the authors' response that the 20th, 40th, 60th, and 80th percentiles correspond to quartiles is statistically inaccurate (quartiles are the 25th, 50th, and 75th percentiles). The use of the 20th, 40th, 60th, and 80th percentiles effectively divides the data into five groups, which contradicts the stated goal of "classifying countries into four exposure levels." Please provide a clear and correct justification for the chosen grouping method.

Furthermore, should other datasets beyond the poverty data also suffer from timeliness issues, the discussion section should be expanded to address the implications of these limitations for the proposed framework.

The authors are encouraged to incorporate relevant recent literature, such as the following studies in China, to further strengthen the study's justification and discussion:

doi: 10.1016/S2468-2667(22)00256-0

doi: 10.1016/j.glohj.2021.08.001

doi: 10.1186/s12939-019-0982-6

I recommend acceptance pending satisfactory revision of this point.

7. PLOS authors have the option to publish the peer review history of their article (what does this mean?). If published, this will include your full peer review and any attached files.). If published, this will include your full peer review and any attached files.

**Do you want your identity to be public for this peer review?** For information about this choice, including consent withdrawal, please see our Privacy Policy..

Reviewer #1: No

Figure Resubmissions:

---

## [Decision Letter · Decision Letter 2]

25 Mar 2026

Measuring Health Financing Vulnerability Due to Reductions in Official Development Assistance: A Conceptual Framework with Empirical Application Across 47 African Countries

PGPH-D-25-02231R2

Dear Prof Asamani,

We are pleased to inform you that your manuscript 'Measuring Health Financing Vulnerability Due to Reductions in Official Development Assistance: A Conceptual Framework with Empirical Application Across 47 African Countries' has been provisionally accepted for publication in PLOS Global Public Health.

Best regards,

Muhammad Iqhrammullah, Ph.D

Academic Editor

Reviewer Comments (if any, and for reference):

Reviewer's Responses to Questions

**Comments to the Author**

1. If the authors have adequately addressed your comments raised in a previous round of review and you feel that this manuscript is now acceptable for publication, you may indicate that here to bypass the “Comments to the Author” section, enter your conflict of interest statement in the “Confidential to Editor” section, and submit your "Accept" recommendation.

Reviewer #1: All comments have been addressed

2. Does this manuscript meet PLOS Global Public Health’s publication criteria? Is the manuscript technically sound, and do the data support the conclusions? The manuscript must describe methodologically and ethically rigorous research with conclusions that are appropriately drawn based on the data presented.? Is the manuscript technically sound, and do the data support the conclusions? The manuscript must describe methodologically and ethically rigorous research with conclusions that are appropriately drawn based on the data presented.

Reviewer #1: Yes

3. Has the statistical analysis been performed appropriately and rigorously?

Reviewer #1: Yes

4. Have the authors made all data underlying the findings in their manuscript fully available (please refer to the Data Availability Statement at the start of the manuscript PDF file)?

The PLOS Data policy requires authors to make all data underlying the findings described in their manuscript fully available without restriction, with rare exception. The data should be provided as part of the manuscript or its supporting information, or deposited to a public repository. For example, in addition to summary statistics, the data points behind means, medians and variance measures should be available. If there are restrictions on publicly sharing data—e.g. participant privacy or use of data from a third party—those must be specified.requires authors to make all data underlying the findings described in their manuscript fully available without restriction, with rare exception. The data should be provided as part of the manuscript or its supporting information, or deposited to a public repository. For example, in addition to summary statistics, the data points behind means, medians and variance measures should be available. If there are restrictions on publicly sharing data—e.g. participant privacy or use of data from a third party—those must be specified.

Reviewer #1: Yes

5. Is the manuscript presented in an intelligible fashion and written in standard English?

Reviewer #1: Yes

6. Review Comments to the Author

Reviewer #1: The authors have adequately addressed all of my concerns.

7. PLOS authors have the option to publish the peer review history of their article (what does this mean?). If published, this will include your full peer review and any attached files.). If published, this will include your full peer review and any attached files.

**Do you want your identity to be public for this peer review?** For information about this choice, including consent withdrawal, please see our Privacy Policy..

Reviewer #1: No
